# The Spanish Olive Oil with Quality Differentiated by a Protected Designation of Origin

Ana García-Moral, Encarnación Moral-Pajares  and Leticia Gallego-Valero *

Department of Economy, University of Jaén, 23071 Jaén, Spain; agm00037@red.ujaen.es (A.G.-M.);
emoral@ujaen.es (E.M.-P.)
* Correspondence: lgallego@ujaen.es

**Abstract:** The Protected Designation of Origin (PDO), part of the EU's quality policy for agri-food products, aims to provide consumers with reliable information on the quality of a food, linked to its origin. Olive oil has perceptible qualities derived from its place of production, which create a link between the product and its place of origin, and which can influence consumer preferences. Spain, the world's leading producer of this vegetable fat, had 29 PDOs at the end of 2020, 25.84% of the EU total for this industry. Based on the arguments drawn from the literature and the information provided by the Spanish Ministry of Agriculture, Fisheries and Food (MAPA), this paper first analyses the importance of olive oil with differentiated quality certified by a PDO for the Spanish olive oil industry. Secondly, the *t*-test is applied to identify positive differences in the income earned by farmers who produce olive oil certified by a PDO. Thirdly, the international competitiveness of extra virgin olive oil (EVOO) bearing a PDO label is analysed using the Revealed Comparative Advantage (RCA) index. The evidence confirms that PDO certification adds value to the product and promotes exports. However, the Spanish olive oil industry does not perform well enough to harness the potential offered by this quality label, it as it does not manage to sell all the PDO-certified EVOO. This situation merits further investigation in future studies, and should be taken into account in the design of actions and campaigns organised by institutions involved in the industry. This article contributes to the evaluation of the quality policy for EU agri-food products and examines the recent evolution of the Spanish PDO-certified olive oil industry.

**Keywords:** extra virgin olive oil (EVOO); Protected Designation of Origin (PDO); Spain; export; Revealed Comparative Advantage

## 1. Introduction

Daily consumption of extra virgin olive oil (EVOO) has important health benefits, thanks to its cardioprotective, antioxidant, anti-inflammatory, and antitumorigenic properties [1–4]. Spain, located on the western edge of the Mediterranean and with 2347 million hectares of olive groves dedicated to the cultivation of olives for oil, has long been the world's leading producer of this vegetable fat [5]. In the 2020–2021 season, its production was estimated at 1389 thousand tonnes, accounting for 46% of global olive oil production (Figure 1), according to data published by the International Olive Council (IOC) [6]. It was followed by Greece, with 275 thousand tonnes (9.11%) and Italy with 273 thousand tonnes (9.11%). At the same time, it is the Mediterranean countries that have the highest annual consumption per inhabitant, according to the IOC, with values of 11.4 kg/inhabitant in Spain, 10.3 in Greece, 7.1 in Italy, 6.3 in Cyprus, 6.8 in Syria, and 5.8 kg/inhabitant in Portugal. Compared to these figures, consumption is far lower in countries outside the Mediterranean arc, such as Switzerland, the United States, or Canada, with values of 2.1, 1.2, and 1.5 kg/inhabitant, respectively. This situation is due to various factors such as cultural differences, personal tastes and traditions, education or geographical proximity between regions that influence food preferences and food availability [7,8]. Nevertheless,

there is a growing demand, bolstered by public and private promotional campaigns in non-traditional markets, which have helped to stimulate interest in the consumption of this food, for its health benefits [9].

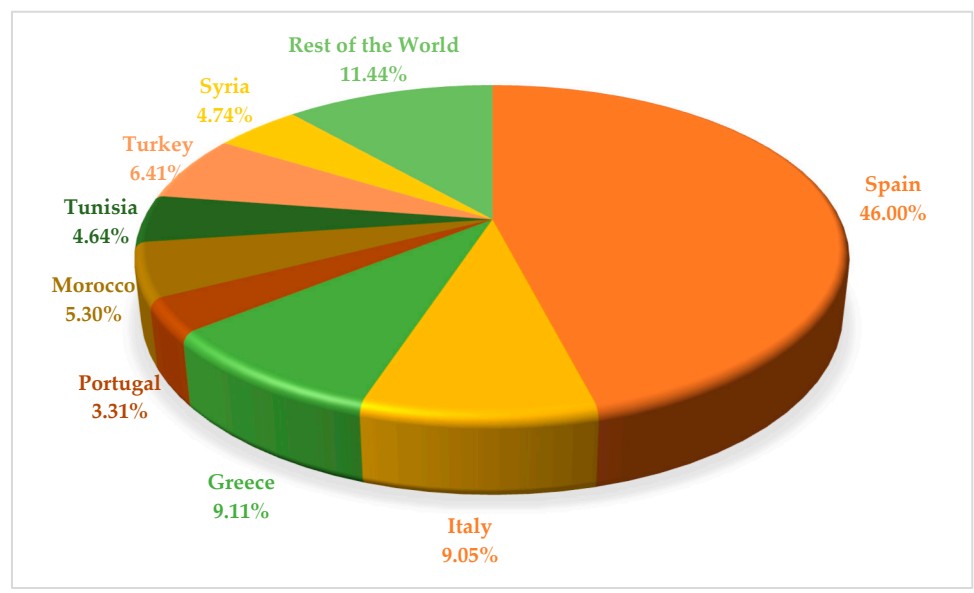

**Figure 1.** Distribution of global olive oil production by country in the 2020–2021 season. Source: [6].

Spain has been a pioneer in the regulation of differential quality linked to a place of origin. The 1932 Wine Statute established the Designation of Origin as the geographical label used in national and foreign markets to designate wines that traditionally have special production characteristics and use wine making and ageing processes typical of the district or region from which they take their name. Spain's entry into the EU in 1986 entailed its acceptance of the acquis communautaire and the recognition of the supremacy of EU laws over domestic ones. Similar to Spain, many other countries have attempted to develop rules and regulations governing the use of the name of a country, a region or a specific place to designate a product originating there, whose quality or characteristics are essentially due to the geographical environment, including natural and human factors [10].

According to Regulation (EU) No. 1151/2012 of the European Parliament and of the Council of 21 November 2012 on quality schemes for agricultural products and foodstuffs, the label PDO "*identifies a product: (a) originating in a specific place, region or country; whose quality or characteristics are essentially or exclusively due to a particular geographical environment with its inherent natural factors; and the production steps of which all take place in the defined geographical area*". In Spain, the Regulatory Council of the PDO is responsible for guaranteeing the authenticity and traceability of EVOO with a PDO, through the certification of the product by a control body accredited by the National Accreditation Entity (ENAC) designated by the Government under Regulation (EC) No. 765/2008.

The goal of this European provision is to harmonise the legal framework established thus far in the member states, helping to protect against fraud through a common, institutionalised policy [11]. It also aims to help consumers avoid the confusion and distrust which could be caused by diverging criteria in multiple legal provisions [12]. In this regard, by establishing intellectual property rights, the products listed on the legal register "eAmbrosia" [13] are legally protected against fraud and imitations within the EU and in third countries that have signed specific agreements; as a result, member states must take the necessary measures to protect registered names, and to prevent and block their illicit use [14].

The PDOs for agricultural products and foodstuffs established by the EU are unevenly distributed by type of product and country, depending on territorial factors conditioned by socioeconomic and sociocultural elements [15,16]. In the eAmbrosia database, there are

659 PDOs (excluding wine and spirits) with a registration date prior to 31/12/2020. Of these PDOs, 622 are from one of 24 of the EU-27 countries, since Denmark, Estonia, and Malta have not registered any. In addition, five non-EU countries have PDOs recognised by the EU, after having complied with all the controls and assessments associated with European standards and having passed the European Commission's approval procedure. They are the Dominican Republic (1), China (4), Turkey (4), the United Kingdom (27), and Vietnam (1). The olive oil industry has 112 recognised PDOs as of 31/12/2020, representing 16.99% of the total registered in the EU. As such, the industry ranks third after cheese, with 195 (29.59%), and fresh or processed cereals, fruits and vegetables, with 160 (24.28%). By country, Italy has the most certifications in the olive oil industry as of 31/12/2020, with 42 (37.50%), followed by Spain with 29 (25.89%), shown in Figure 2, Greece with 20 (17.85%), and France with 8 (7.14%).

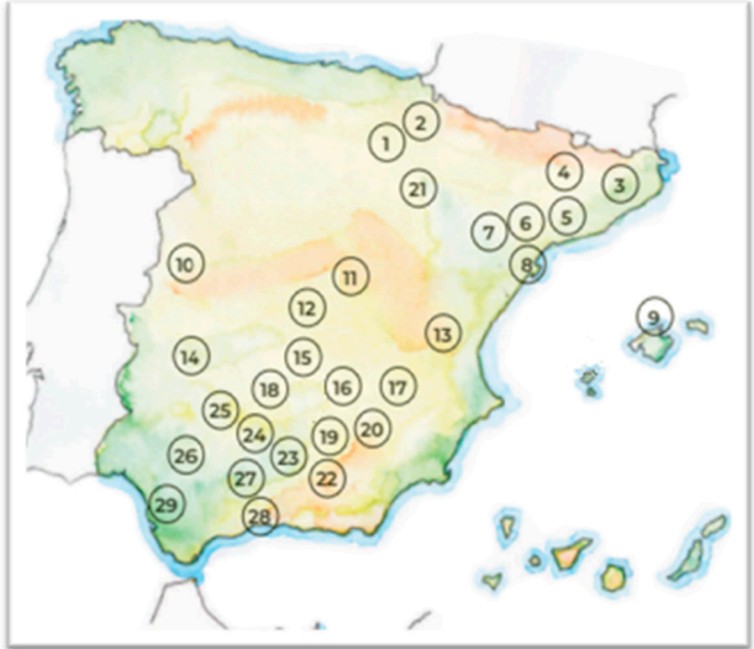

| | |
|---|---|
| 1. Aceite de La Rioja | 16. Aceite Campo de Montiel |
| 2. Aceite de Navarra | 17. Sierra de Segura |
| 3. Aceite de L'Empordà | 18. Montoro-Adamuz |
| 4. Les Garrigues | 19. Sierra Mágina |
| 5. Siurana | 20. Sierra de Cazorla |
| 6. Aceite de Terra Alta | 21. Aceite Sierra del Moncayo |
| 7. Aceite del Bajo Aragón | 22. Montes de Granada |
| 8. Aceite del Baix Ebre-Montsià | 23. Poniente de Granada |
| 9. Aceite de Mallorca | 24. Priego de Córdoba |
| 10. Gata-Hurdes | 25. Baena |
| 11. Aceite de La Alcarria | 26. Estepa |
| 12. Montes de Toledo | 27. Aceite de Lucena |
| 13. Aceite de la Comunitat Valenciana | 28. Antequera |
| 14. Aceite Monterrubio | 29. Sierra de Cádiz |
| 15. Aceite Campo de Calatrava | |

**Figure 2.** PDO-certified EVOO from Spain. Source: [17].

Based on the arguments presented above, and in order to advance the knowledge of the Spanish olive oil industry and assess its commitment to quality differentiated by a PDO, the following research questions are posed:

Q.1. What share of EVOO production in Spain (the country responsible for 46% of global olive oil production in the 2020/21 season) is PDO certified?
Q.2. Is the mean value per kilogram of the PDO-certified EVOO (EVOO-PDO) sold by Spanish companies higher than that of non-PDO-certified EVOO?
Q.3. Does Spanish EVOO-PDO have a comparative advantage, contributing to the competitiveness of the industry as a whole in the international market?

To date, most of the studies focusing on EVOO-PDO in Spain, Italy, Greece, or Turkey have analysed specific cases [18–25]. Only the study by [26] addresses the Spanish olive oil industry as a whole. The present study offers an aggregate view of the agroindustry that produces PDO-certified olive oil in the country that is the world's leading supplier of this foodstuff. From the results, conclusions can be drawn about the entire industry, which will be useful for decision making by national and supranational institutions, such as the national government or the European Commission, whose actions and policies are aimed at the industry as a whole rather than specific cases. By conducting a rigorous analysis of the available empirical data, this article seeks to contribute to the assessment of the EU

agri-food products quality policy. Moreover, it identifies the main features of the recent evolution of the Spanish EVOO-PDO industry.

## 2. Materials and Methods

### 2.1. Contextual Framework

A geographical indication (GI) is a label used on products that have a specific geographical place of origin and possess qualities or a reputation specifically stemming from that place of origin, as established in Article 22.1 of the Agreement on Trade-Related Aspects of Intellectual Property Rights (TRIPS Agreement), adopted in 1994 [10]. Olive oil has qualities that are derived from the place where it is produced. It is shaped by specific local factors such as the variety of olives cultivated for oil in the territory, the system used to produce the oil, the climate, or the type of terrain. This link between the product and its place of origin can be perceived in its organoleptic properties and can influence consumer preferences [27–30], as is the case with other foods [31–33].

The main advantage of the system for the assessment and protection of PDO, Protected Geographical Indication (PGI) and Traditional Specialities Guaranteed (TSG), created by the EU in 1992 before being revised in 2006 (Council Regulation (EC) No. 510/2006 of 20 March 2006 on the Protection of Geographical Indications and Designations of Origin for Agricultural Products and Foodstuffs) and 2012, is that the production standards or parameters are published in a single harmonised register. It has also successfully established a procedure for inspections by external certification bodies. The corresponding label or seal, which certifies the quality of a product and draws consumers' attention to it, is the result of a process supported by a network of governmental and non-governmental institutions that verifies the real origin of the product, the specific raw materials used, or the traditional manufacturing technique applied in a specific territorial area [34,35]. This certification provides producers with an exclusive right to use the label, in principle giving them a clear advantage over others [36].

The image of product quality, assured under the PDO/PGI symbol, can foster consumers' preference for the product and increase their willingness to pay, especially in local markets [37,38]. This raises the value of the product and boosts producers' incomes [39]. Specifically, in the case of olive oil, various authors [18,40] have concluded that its origin influences consumers' choice of the producing region and country and their willingness to pay, thus supporting the competitiveness of the product. Furthermore, the quality of the product, as indicated by its PDO/PGI certification, can help boost sales in international markets [41–46]. The European Commission's 2021 report [47] concludes that the value of the certified agricultural products and foodstuffs was on average twice that of the sales of similar products without this certification.

However, there are notable differences by product types and countries [48–50]; although PDOs/PGIs can generate added value, especially at the consumer and retail level, the positive effects on producers' incomes are not as clear. According to [51], in the olive oil industry, the price charged by the producer may not be enough to offset the costs of adopting the protection regime. One reason for this could be that the farmer, an EVOO producer under a PDO/PGI, does not directly reach the end consumer; in longer supply chains it is the retailers/distributors who benefit from much of the added value of the foodstuff [50,52]. This situation discourages producers in the EVOO-PDO value chain as they bear the certification costs [53].

### 2.2. Data and Methodology

The empirical analysis conducted here uses secondary information from different national institutions such as the Spanish Ministry of Agriculture, Fisheries and Food (MAPA) [54] and the Secretariat of State for Trade of the Ministry of Industry, Trade and Tourism of Spain. The period under analysis is determined by the publication of data on the different variables: number of PDOs, number of industries, production, economic value of production, and export. Since 2008, MAPA's Division for Differential Quality and Organic

Agriculture, in collaboration with the Regulatory Councils of Protected Designations of Origin and Protected Geographical Indications and associated entities, has published the report "Data on Protected Designations of Origin (PDOs) and Protected Geographical Indications (PGIs) of Agri-Food Products" [54]. In the case of EVOO, there is no recognised PGI in Spain that produced any oil during the analysed period; therefore, the study focuses on the productive and commercial activity of the PDOs between 2008 and 2020, both included.

It should be noted that the information recorded in official statistics is sometimes incomplete or based on estimates [52]. Different studies carried out by [47,55] show that there is limited available data for evaluating the effectiveness of the PDO/PGI in the EU. Furthermore, the following clarifications should be taken into account:

(a)   The classification of virgin olive oil (VOO) established by the European Commission is used [56] based on quality parameters related to: (i) physico-chemical characteristics, such as the acidity level, peroxide index, fatty acid content, and sterol composition; (ii) organoleptic (sensory) characteristics, such as the fruitiness and the absence of organoleptic defects. The classification differentiates between:

  1.   EVOO, which is the highest quality category.
  2.   VOO, which may have some sensory defects.
  3.   Lampante virgin olive oil (LVO), a lower quality virgin olive oil that is refined or used for industrial purposes.

They are all produced using mechanical processes (washing, decanting, centrifuge, and filtering) under specific thermal conditions that do not cause any alteration in the product [1]. According to the specifications for the 29 Spanish olive oil PDOs published on the EU website, the product certified by the PDO Regulatory Councils must be EVOO.

(b)   The area covered with olive groves that cultivate olives for milling to produce EVOO-PDO must be registered in a PDO. The Regulatory Council of the PDO is the body responsible for verifying the origin, traceability and quality of EVOO, and ensuring that it is produced in accordance with the specifications of the PDO in question, approved by the European Commission under Law 6/2015, of 12 May. The product certified by the Regulatory Council, produced in its entirety in a specific geographical area, is identified with the label shown in Figure 3.

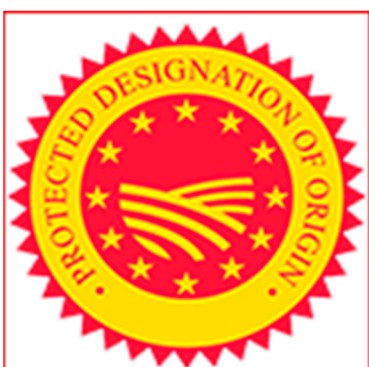

**Figure 3.** EU Protected Designation of Origin label. Source: [57].

(c)   The information provided by the Datacomex database of the Secretariat of State for Trade of Spain's Ministry of Industry, Trade and Tourism for the period 2008–2016 lists, in accordance with the EU TARIC nomenclature in effect in those years, identifies VOO under the tariff subheading 1509 10—differentiating between 1509 10 10, LVO and 1509 10 90, VOO, and EVOO. As of January 2017, according to Commission Implementing Regulation (EU) 2016/1638 of 6 September 2016, subheading 1509 10, VOO includes 1509 10 10, LVO; 1509 10 20, EVOO; 1509 10 80, other VOO. Between the years 2017 and 2020, 86% of the total amount of VOO exported by Spain (identified as

1509 10) was EVOO (subheading 1509 10 20). This figure has been used to estimate the value of EVOO exports from Spain in the period 2008–2016.

(d) The price charged by the farmer at source for the olive is what it is received for the product delivered to the olive oil mill. The olive oil mill, which is responsible for milling the fruit, and usually for bottling and marketing, pays the farmer a price depending on the yield of the olive obtained and the quality of the fruit. The quality of the fruit determines whether or not EVOO-PDO can be produced [58]. The highest quality olives should be picked from the tree at the beginning of the harvest season, when the fruit has a greenish-purple skin [59], but the yield of the fruit is lower. This means higher production costs, since more kilos of fruit are needed to produce a kilo of oil.

(e) Exports of agri-food products from Spain are measured by the amount in euros of the registered sales, in accordance with the TARIC tariff classification, chapters 1 to 24 (Council Regulation EEC No. 2658/87).

Table 1 shows the variables used in the statistical tests, the units of measurement and the sources of information.

**Table 1.** Variables, unit of measurement, and sources.

| Variables | Unit of Measurement | Sources |
|---|---|---|
| Surface area of olive groves dedicated to the production of EVOO-PDO in Spain (EVOO-PDO Surface) | Hectares (ha) | SGAA |
| Production of EVOO-PDO in Spain (P EVOO-PDO) Production of EVOO-PDO in Spain sold on the market (S EVOO-PDO) Value of the EVOO-PDO produced in Spain sold on the market (SV EVOO-PDO) | Tonnes (t) | SGAA |
| Value of the PDO-certified agri-food produced in Spain sold on the market (SV Agrifood-PDO) Mean value per kilogram EVOO-PDO (V kg EVOO-PDO) Exports of EVOO-PDO (X EVOO-PDO) Exports of PDO-certified agri-food products from Spain (X Agrifood-PDO) | Euros (€) | SGAA |
| Surface area of olive groves dedicated to the production of virgin olive oil in Spain (Surface) | Hectares (ha) | MAPA ESYRCE |
| Production of EVOO in Spain (P EVOO) | Tonnes (t) | MAPA |
| Mean value per kilogram EVOO (V kg EVOO) | Euros (€) | MAPA |
| Exports of EVOO from Spain (X EVOO) Export of agri-food products from Spain (X Agrifood) | Euros (€) | DATACOMEX |

Source: own elaboration.

A descriptive analysis of the relative importance of PDO-certified olive oil production in Spain is conducted using data extracted from different statistical sources. To check for a difference between the average value per kg of EVOO and that of EVOO-PDO, Student's *t*-test is used, which compares the means of two variables that follow a normal distribution. The analysis of the comparative advantages of the EVOO-PDO industry is based on the Revealed Comparative Advantage (RCA) index [60]. In this case, the adapted RCA index is used, which compares the share of EVOO-PDO exports with the share of EVOO exports in Spain's total agri-food exports. The following formula is used:

$$RCA = \frac{\frac{X\ EVOO-PDO_t}{X\ Agrifood-PDO_t}}{\frac{X\ EVOO_t}{X\ Agrifood_t}}$$

where:

$X\ EVOO\text{-}PDO_t$ = Exports of EVOO-PDO in year t;

$X\ Agrifood\text{-}PDO_t$ = Exports of agrifood-PDO in year t;

X EVOO$_t$ = Exports of EVOO in year t;
X Agrifood$_t$ = Exports of agrifood in year t.

An RCA value equal to or greater than 1 indicates that the industry has a comparative advantage in the export of EVOO-PDO, while RCA values lower than 1 indicate a comparative disadvantage, given the relatively smaller share of EVOO-PDO exports [60]. The RCA is calculated for each year and as the average for the period [61].

## 3. Results

Q.1. What Share of EVOO Production in Spain (the Country Responsible for 46% of Global Olive Oil Production in the 2020/21 Season) Is PDO Certified?

In 2008, the olive grove area in Spain dedicated to the cultivation of olives for milling totalled 2475.36 thousand ha, of which 958.42 thousand ha, or 38.97%, was dedicated to the production of EVOO-PDO. Over the course of the analysed period, the area of land associated with PDO-certified production decreased, dropping to 710.32 thousand ha in 2020, or 27.81% of the total (Table 2). In these years, however, the volume of PDO-certified production increased, as happened in the Spanish olive oil industry as a whole, due to technical improvements applied in the preceding decade and the increase in the irrigated area, boosting productivity overall. In 2020, the Spanish olive oil industry produced 74.36 thousand t of EVOO-PDO, 12.35% of all EVOO produced in this country. However, in seasons when favourable weather conditions led to better-quality olives for milling, the amount of EVOO-PDO exceeded 110 thousand t, representing 38.24% of the total produced in 2012 [62]. Therefore, the answer to the first research question posed (Q.1) is that EVOO certified by a PDO in Spain accounts for a relatively small share of total national olive oil production, with an average share of 15.60% of the total volume of EVOO produced between 2008 and 2020.

**Table 2.** Olive grove area dedicated to the production of EVOO-PDO, production of EVOO-PDO, and sales of EVOO-PDO in Spain between 2008 and 2020.

| | Surface EVOO-PDO | | P EVOO-PDO | | S EVOO-PDO | |
|---|---|---|---|---|---|---|
| | Thousand ha | % Total Area Dedicated to the Cultivation of Olives for Milling | Thousand t | % Total Production of EVOO | Thousand t | % Total P EVOO-PDO |
| 2008 | 958.42 | 38.97 | 48.61 | 10.27 | 27.63 | 56.84 |
| 2009 | 1113.69 | 44.99 | 83.90 | 17.27 | 25.46 | 30.34 |
| 2010 | 703.50 | 28.72 | 99.99 | 18.00 | 22.14 | 22.14 |
| 2011 | 705.34 | 28.87 | 124.66 | 17.72 | 25.46 | 20.42 |
| 2012 | 657.48 | 26.95 | 136.01 | 38.24 | 26.07 | 19.17 |
| 2013 | 696.15 | 28.59 | 132.98 | 16.16 | 25.95 | 19.51 |
| 2014 | 688.25 | 28.14 | 144.42 | 33.71 | 29.20 | 20.22 |
| 2015 | 653.73 | 26.62 | 72.00 | 7.78 | 27.67 | 38.44 |
| 2016 | 694.73 | 28.12 | 90.50 | 10.92 | 28.90 | 31.93 |
| 2017 | 700.39 | 28.06 | 83.08 | 11.38 | 28.28 | 34.04 |
| 2018 | 710.11 | 28.29 | 79.16 | 9.72 | 30.67 | 38.74 |
| 2019 | 709.50 | 27.89 | 112.76 | 22.67 | 30.32 | 26.89 |
| 2020 | 710.33 | 27.81 | 74.36 | 12.35 | 32.05 | 43.10 |
| Average 2008–2020 | 746.28 | 30.15 | 98.65 | 15.60 | 27.68 | 28.05 |

Source: [63].

On the other hand, the data in Table 2 confirm that the supply of EVOO-PDO far exceeds demand. The average volume sold in the period 2008–2020 is only 27.68 thousand t, with a maximum value of 32.05 thousand t in 2020, or 28.05% of the total produced.

Figure 4 shows the percentage share of the sales value of EVOO-PDO in total revenues from the sale of PDO-certified agri-food products in Spain between 2008 and 2020. Up until

2015, the olive oil industry was responsible for more than 10% of all revenues from the sale of PDO-certified agri-food products. However, between 2015 and 2020, many types of agri-food products registered significant increases in sales as a result of a greater commitment to differential quality and consumers' growing interest in this type of food. Examples include PDO-certified ham, fruit, and fish and molluscs, which registered increases in sales revenues of 29.68%, 121.97%, and 58.57%, respectively, between 2015 and 2020. In comparison, revenues from EVOO-PDO rose by 6.46%, with this increase being below the average for the whole sector (SV Agrifood-PDO), which presented an increase of 32.89%. This trend coincides with what has been observed in the EU as a whole [47] and with the increase in the production and sale of PDO-certified fruit and vegetable products in Spain [64].

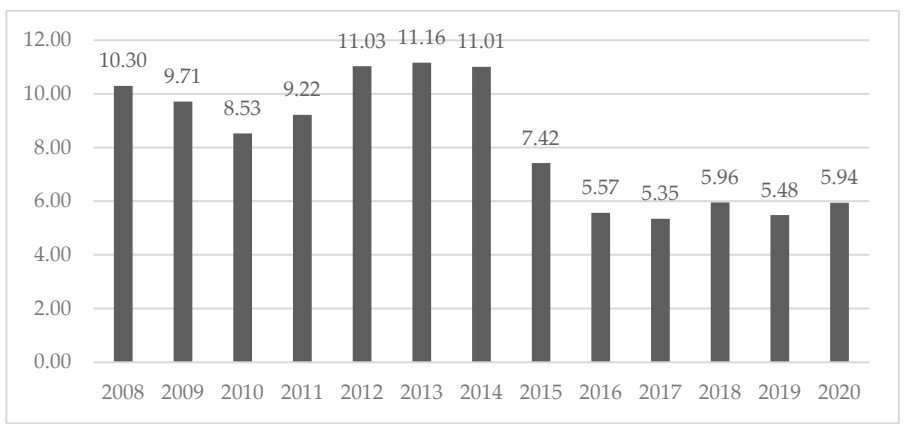

**Figure 4.** Percentage share of SV EVOO-PDO in SV Agrifood-PDO between 2008 and 2020. Source: [54].

Q.2. Is the mean value per kilogram of the PDO-certified EVOO (EVOO-PDO) sold by Spanish companies higher than that of non-PDO-certified EVOO?

To understand whether the effort made by the Spanish olive oil industry with differential quality corresponds to higher income for the farmer, as in the second research question (Q.2) posed, the average value per kg of EVOO earned by farmers each year is compared with the corresponding value for EVOO-PDO [63]. To produce a superior quality olive oil, the farmer must bring forward the harvest [59], which negatively affects the yield of the fruit and means that more kilos of olives must be harvested to produce a kilo of EVOO. The olive oil mill, which is responsible for milling the fruit, and usually for bottling and marketing, pays the farmer a price per kilo of EVOO-PDO produced from the olives harvested by the farmer.

Table 3 shows the results obtained with the *t*-test based on data from 2008 to 2020, which allows for confirming whether there are significant differences between the variables compared. This parametric test is used when the variables are normally distributed and homoscedastic. To verify these conditions, the Shapiro–Wilk and Shapiro–Francia tests are carried out, which indicate that there is normality at a confidence level of 90%, and the White and Breusch–Pagan tests, which indicate that there is homoscedasticity at a confidence level of 90%.

The results of the *t*-test can be seen in Table 3. At a confidence level of 95%, there is a significant difference between the two variables, with the value per kg of EVOO-PDO being higher on average. In 2020, the average value received by the farmer per kg of EVOO was EUR 2.15, while the PDO-certified product had a mean value of EUR 4.54, offsetting the fruit yield losses of early harvesting.

**Table 3.** Results of the *t*-test.

|  | EVOO | EVOO-PDO |
|---|---|---|
| Mean | 2.57 | 4.23 |
| Variance | 0.36 | 0.36 |
| Observations | 13 | 13 |
| Pooled variance | 0.37 | |
| Hypothesised mean difference | 0 | |
| Degrees of freedom | 24 | |
| t statistic | −6.99 | |
| P (T <= t) one tail | 0.000000156 | |
| t critical (one tail) | 1.71 | |
| P (T <= t) two tail | 0.000000312 | |
| t critical (two tail) | 2.06 | |

Source: [54,63].

Q.3. Does Spanish EVOO-PDO Have a Comparative Advantage, Contributing to the Competitiveness of the Industry as a Whole in the International Market?

To answer the third research questions (Q.3), Table 4 shows the sales of EVOO-PDO in the international market as a share of total sales of PDO-certified agri-food products, the percentage share of EVOO exports in the total revenue from Spain's agri-food trade balance, and the RCA value. First of all, the data corroborate the outward facing nature of PDO-certified olive oil production, which in 2020 accounted for 8.90% of all PDO-certified agri-food exports. This percentage far exceeds the weight of this industry in the sales value of Spanish PDO-certified agri-food production, which was 5.94% in that year [54]. It is also greater than the share of EVOO exports in the total revenue from the sale of agri-food products on the international market, which represented 3.71% in 2020. Secondly, it can be observed that the PDO-certified olive oil industry shows a notable specialisation in exports, given that the RCA registered a value above 1 every year, reaching 3.55 in 2012. Between 2018 and 2020, exports of EVOO-PDO tripled in value, registering an increase of more than EUR 26 million in this period. A drop in exports is only observed in years of poor harvests, with lower fruit quality, such as 2016 and 2017.

**Table 4.** Share of EVOO-PDO exports in Spain's total exports of Agrifood-PDO, share of EVOO exports in Spain's total exports of agri-food, and RCA between 2008 and 2020.

|  | X EVOO-PDO X Agrifood-PDO (%) | X EVOO X Agrifood (%) | RCA |
|---|---|---|---|
| 2008 | 5.69 | 4.38 | 1.30 |
| 2009 | 8.60 | 3.86 | 2.22 |
| 2010 | 8.27 | 4.40 | 1.88 |
| 2011 | 13.61 | 3.94 | 3.46 |
| 2012 | 12.89 | 3.63 | 3.55 |
| 2013 | 12.10 | 3.66 | 3.31 |
| 2014 | 15.51 | 4.85 | 3.20 |
| 2015 | 12.26 | 4.19 | 2.92 |
| 2016 | 8.40 | 4.89 | 1.72 |
| 2017 | 8.31 | 5.03 | 1.65 |
| 2018 | 9.20 | 4.21 | 2.19 |
| 2019 | 8.86 | 4.09 | 2.17 |
| 2020 | 8.90 | 3.71 | 2.40 |
| Total 2008–2020 | 9.92 | 4.23 | 2.35 |

Source: [54,65].

## 4. Discussion

Between 2008 and 2020, the sales of S EVOO-PDO followed a rising trend, registering an increase of 15.99%, with a 43.78% increase in its SV EVOO-PDO, reaching EUR

145.49 million in 2020. However, this rise is less marked than in the Spanish PDO-certified agri-food industry as a whole, which registers an increase in sales of 146.18%, reaching EUR 2.45 billion in 2020. Furthermore, it is noteworthy that the Spanish olive oil industry never managed to place the entire PDO-certified output on the market, and more than 40% of the certified production was directed through traditional EVOO sales channels. Indeed, there were very few years in which the sale of the PDO-certified product exceeded 30% of the total. The analysis carried out confirms that the effort made by producers to offer an olive oil of certified quality does not correspond to the capacity of the market to absorb the entire supply. This situation reveals certain weaknesses in the product marketing system, which warrants further analysis. According to the literature reviewed, this problem partly reflects the lack of consumer awareness about the additional value that a PDO label implies, as well as failings on the supply side relating to the fragmentation of production and the negotiating power of large brands [63]. From 2015 onwards, a third of the volume of PDO-certified produced was sold, reaching a maximum in 2020, a year in which the market absorbed 32.05 thousand t of EVOO-PDO, 43.10% of the total produced. These results reflect the increased interest in olive oil in the international market [9], and more specifically, certain consumer groups' growing preference for certified EVOO over the last decade [18,66].

It has been assumed that the Spanish farmer, a producer of quality olives, obtains a premium price for the product he/she sells to produce PDO-certified EVOO. The results align with those from other studies which find that information about the origin of the product and the production process certified by a PDO increases consumers' willingness to pay more for olive oil [39] and for food in general [67]. However, in some cases, the PDO does not influence the price [68]; indeed, there are important differences across countries and consumer reactions vary widely [48,69].

Spain enjoys notable comparative advantages in the production of EVOO. The sale on the international market of this vegetable fat makes a substantial contribution to the agri-food trade balance of the Spanish economy. In 2008, the estimated revenues from the export of EVOO amounted to EUR 1.21 billion, representing 4.38% of total revenue in the country's agri-food trade balance. In this context, it is worth analysing whether the PDO-certified product has contributed to these results, presenting better export performance than the average for the sector. These results reflect the greater efficiency shown by the Spanish PDO-certified olive oil industry that sells part of its output in international markets [26] and support the findings reported by [41,44,70] in the case of wine.

## 5. Conclusions

Using MAPA data on the production and sales of EVOO-PDO, this study contributes to the literature that analyses the effectiveness of the PDO as a differential quality instrument promoted by the EU. The PDO aims to ensure farmers and producers a fair income for a higher quality product, by providing clear information on the specific product characteristics linked to its geographical origin, and by providing consumers with valuable information for their purchase process.

By conducting an empirical and descriptive–reflective analysis, the purpose is to determine the importance of PDO-certified olive oil production in relation to the Spanish olive oil industry as a whole.

It also analyses whether farmers earn a higher income for a better quality, PDO-certified product, as indicated in the preamble of Regulation (EU) No. 1151/2012 on the quality regimes of agricultural and food products. Finally, the competitiveness of EVOO in the international market is examined.

Based on the results of the study, questions posed in the introduction can be answered. First of all, it can be confirmed that there is a growing demand for PDO-certified agri-food products from Spain. Between 2008 and 2020, PDO-certified EVOO sales registered a 15.99% increase in volume, compared to a rise in global olive oil consumption of 7.68%, according to the IOC. In 2020, the industry placed 32,049.08 tonnes of PDO-certified EVOO

on the market, with a total value of EUR 145.49 million. However, despite the rising trend in consumption, which explains the continued increase in sales, not all the certified output is sold as such: a high percentage of the oil produced under the quality standards required by a PDO are marketed simply as EVOO. Despite this, in recent years, especially following the outbreak of the pandemic, there has been growing concern about health, which has encouraged the consumption of good quality foods [9]. PDO-certified EVOO is just such a food, which partly explains the maximum value reached for this product in 2020. That said, the fact that this market segment is unable to absorb all the high quality, PDO-certified oil produced calls for a more in-depth analysis in future research.

Secondly, it is shown that farmers who produce higher quality olives, allowing them to obtain PDO-certified EVOO, are rewarded with higher earnings for their harvest, which has a positive impact on their income. These results reflect consumers' willingness to pay more for a quality product with geographically-specific attributes, as different studies have shown for both the olive oil industry and for food in general [39,67].

Finally, in line with the marked external orientation of Spanish olive oil industry—every year it places two thirds of its total production in international markets—strong sales of PDO-certified olive oil are observed on international markets, confirming the growing interest of foreign customers in this foodstuff. It also demonstrates the positive contribution of this product to the competitiveness of the national olive oil industry as a whole and to the export revenues in the country's trade balance.

However, the Spanish olive oil industry's clear commitment to quality production and PDO certification requires an appropriate marketing strategy to guarantee that everything that is produced can be sold. Being more market oriented would allow farmers linked to a PDO to secure higher revenues for the product, boosting income in the rural areas in which the farms are located. First and foremost, European, national, and regional institutions must lend their support to information and communication campaigns aimed at consumers to raise their awareness and appreciation of the product, both in the national market and beyond. Efforts should also be made to develop the export activity of this agroindustry, with appropriate information, training, and promotion programmes that can be conducted in collaboration with the different institutions involved in the industry. Achieving this would bolster international consumers' growing interest in this high-quality vegetable fat, as reflected by the fact that exports represented 27.17% of the total sold in 2020, compared to 12.68% in 2008. Similarly, there is an urgent need to promote the development of the online sales channel, which shortens the distribution chain and strengthens the relationship between the origin of the product and the final consumer, as some studies have shown for the olive oil industry in southern Spain [71] and the organic olive oil subsector [72,73].

This study does have some limitations. The first is due to the process of aggregating the data on the 29 PDOs published by MAPA; although it allows for an analysis of the entire industry, it does not account for particular aspects of the different PDOs. These differences can be explored in subsequent research. Secondly, the dynamics described above have been influenced by different circumstances. On the one hand, there are the tariffs on Spanish olive oil imposed by the US administration in October 2019, which negatively impacted sales of PDO-certified EVOO to the US, a country that received 14.78% of the total volume exported from Spain in that year. On the other hand, there were the restrictions on international trade caused by the COVID-19 pandemic, which negatively affected exports. Despite these limitations, this work opens up the field for future research on the production of PDO-certified olive oil in the world's leading supplier of this foodstuff. In this respect, in future studies, it would be worth specifying the destination of exports and examining export dynamics by country; a questionnaire could be developed to identify the characteristics of the companies that produce and sell PDO-certified EVOO, the distribution channels they use, their export orientation, and notable characteristics of consumers who buy the product, among other aspects.

**Author Contributions:** Conceptualization, E.M.-P. and A.G.-M.; methodology, A.G.-M.; validation, E.M.-P. and L.G.-V.; formal analysis, A.G.-M.; investigation, A.G.-M.; resources, A.G.-M. and E.M.-P.; data curation, A.G.-M.; writing—original draft preparation, A.G.-M.; writing—review and editing, E.M.-P. and L.G.-V.; supervision, E.M.-P.; project administration, E.M.-P. All authors have read and agreed to the published version of the manuscript.

**Funding:** This research was funded by the IEG (Instituto de Estudios Giennenses, Diputación Provincial de Jaén) 2022, and taking into account the E.M.P. membership to the "Instituto Universitario de Investigación en Olivar y Aceites de Oliva".

**Institutional Review Board Statement:** Not applicable.

**Data Availability Statement:** Data is contained within the article.

**Conflicts of Interest:** The authors declare no conflict of interest.

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
