# Peer review of "The Spanish Olive Oil with Quality Differentiated by a Protected Designation of Origin"

_agriculture, doi:10.3390/agriculture13112169_

Round 1
Reviewer 1 Report
Comments and Suggestions for Authors
Please check the details in the attachment.

Moderate editing of the English language is required.
Author Response
We would like to thank the editor and reviewers for all the comments.
The answers are in the attached file.

Reviewer 2 Report
Comments and Suggestions for Authors
I have read the paper twice, it is very informative but I am unable to grade it as a "research article". Rather, in my opinion it should be published as a "report". Please add a separate literature review and state research hypothesis to make it count as a structured research paper. Please add more references to the discussion section and also create a separate conclusion and implications section. What are theoretical contributions (any underlying theory is missing) of your study? As authors presented single country evidence i.e., Spain, can they further elaborate which cities/regions of the country contribute to the olive production? A region wise bifurcation can be more meaningful if only one country is under consideration.
Comments on the Quality of English LanguageModerate editing of English language required
Author Response
First of all, we would like to thank the editor and reviewers, who have contributed to extensive improvement of the document. The English language have been revised.
In response to the reviewer:
- In research in social sciences, such as agricultural economics, there are many works that start from "research question" and not "hypotheses", which does not diminish the rigor of the analysis carried out.
- The discussion has been separated from the conclusions.
- The analysis carried out has sought to offer an overview of Spanish olive oil with PDO, in which 29 PDOs participate, as represented in figure 1 incorporated into the text. In subsequent work, it is our intention to identify intra-sector differences.
The recomendations of the reviewer have been done.
Reviewer 3 Report
Comments and Suggestions for Authors
General comments
The topic is good, the research problem is exciting and worth analysing. The presentation of the research is clear, reasonable and correct. The language is good, only a few issues of using blue rose or singulars should be checked and corrected occasionally. The various sections are well structured and their content is satisfactory, with very few point to be suggested for correction. These will be mentioned in the following detailed comments. According to the above, I recommend the paper for publication with minor corrections.
Details
Abstract
Around line 16 you mentioned the revealed comparative advantage index as the main analysis tool. Please mention in a few words the other, statistical tools, for example, the T test, applied in the paper.
Introduction
The introduction is clearly written with a good review of relevant literature, about olive oil production, consumption, and about the evolution and importance of the PDO concept. A minor thing: in lines 56 and 64 the format of refering to the EU and EC regulations should be harmonised (the reference to the numbers of the regulations).
In lines 88 to 93 the research questions are described. These are reasonable and precise, however, it would be useful to present them in a numbered form: R1, R2, R3 for easier reference, when later the results are presented.
Materials and methods
This is again a well structured section. Again a few minor comments and suggestions:
In line 157 please add the reference [49] to the report title.
In line 174 the first word ‘The’ should be ‘They’ or ‘These’.
Line 212, Table 1: the heading should clearly mention the names of each column. These should be: name of the variable, explanation, unit of measurement, source of data. That means, you have 4 columns, and each of them should be properly filled for each variable. Currently somewhere the measurement unit is missing, or it is not clear, which sources research which variables.
After Table 1, In line 214, please refer back to the numbered research questions (R1, R2, R3) and indicate how you will evaluate each of them. Currently you give information on R2 in lines 215 and 216, and on R3 in lines 216-230. Please also mention here that the T test as used 4 variables of normal distribution, and mention how you test normality for your variables.
Result
Again a well written section, with a few suggestions for improvement.
The section starts with presenting the answer to the first research question, R1. Please add a subheading (3.1 Result for R1, or something similar).
Lines 246 and 247: actually as Table 2 shows, That decreasing trend is valid only for the years 2008-2010, and the years 2011-2015 are about fluctuations, while the years 2016-2020 reflect an increasing trend. What does the 15.99% actually mean?
Lines 257 and 258: in 2020 the as S EVOO-PDO peaked. Wasn't it negatively affected by the COVID related lockdown?
After line 275 add a sub-section heading (3.2) to refer to answering the R2 research question.
Line 284: to apply the T test the normality of the data should be shown as you also say in line 286. Please give some information about testing the normality of your data.
Before line 293 please add a sub section heading (3.3) reffering to the R3 research question.
Line 311: after”… in 2020.” add, that this tendency prevails in each year from 2008 to 2020.
Discussion and conclusions
This section is generally good, but please reflect briefly on the following issues:
line 350: at the end of the paragraph please add briefly a comparison of the Spanish situation to other empirical results of the same problem for other regions or countries.
Line 356: do we know whether the extra income covers the extra costs for the farmer, in Spain, or in other countries?
With these minor issues, that is, suggested minor revisions, the paper will be good for publication.
References
There are 68 relevant items mentioned, they are up-to-date and give a good overview of the topic.
Comments on the Quality of English LanguageGood, see review above for a few minor issues.
Author Response
We would like to thank the editor and the reviewers for all the comments.
The answers are in the attached file.

Round 2
Reviewer 1 Report
Comments and Suggestions for Authors
Thanks for your revision.
Comments on the Quality of English LanguageMinor editing of the English language is required.
Reviewer 2 Report
Comments and Suggestions for Authors
nill